# Investigating *Candida glabrata* Urinary Tract Infections (UTIs) in Mice Using Bioluminescence Imaging

**DOI:** 10.3390/jof7100844

**Published:** 2021-10-09

**Authors:** Sanne Schrevens, Dominique Sanglard

**Affiliations:** Institute of Microbiology, University of Lausanne and University Hospital, CH-1011 Lausanne, Switzerland; sanne.schrevens@chuv.ch

**Keywords:** *Candida glabrata*, urinary tract infection, bioluminescence

## Abstract

Urinary tract infections (UTIs) are quite common and mainly caused by bacteria such as *Escherichia coli*. However, when patients have urinary catheters, fungal infections comprise up to 15% of these types of infections. Moreover, fungal UTIs have a high mortality, due to rapid spreading of the fungi to the kidneys. Most fungal UTIs are caused by *Candida* species, among which *Candida albicans* and *Candida glabrata* are the most common. *C. glabrata* is an opportunistic pathogenic yeast, phylogenetically quite close to *Saccharomyces cerevisiae*. Even though it is commonly isolated from the urinary tract and rapidly acquires resistance to antifungals, its pathogenesis has not been studied extensively in vivo. In vivo studies require high numbers of animals, which can be overcome by the use of non-invasive imaging tools. One such tool, bioluminescence imaging, has been used successfully to study different types of *C. albicans* infections. For *C. glabrata,* only biofilms on subcutaneously implanted catheters have been imaged using this tool. In this work, we investigated the progression of *C. glabrata* UTIs from the bladder to the kidneys and the spleen. Furthermore, we optimized expression of a red-shifted firefly luciferase in *C. glabrata* for in vivo use. We propose the first animal model using bioluminescence imaging to visualize *C. glabrata* in mouse tissues. Additionally, this UTI model can be used to monitor antifungal activity in vivo over time.

## 1. Introduction

*Candida* species are the most common causative agents of opportunistic, nosocomial, fungal infections [1]. *Candida albicans*, which is well studied, is mostly isolated during vaginal, oral, and invasive infections [2,3,4]. The incidence of other species is on the rise, especially of *Candida glabrata* [1,5]. *C. glabrata* is more closely related to *Saccharomyces cerevisiae* than to *C. albicans* [6,7,8] and is one of the only pathogenic species in this branch of the phylogenetic tree [6,9]. *C. glabrata* lost multiple genes, including complete metabolic pathways, such as nicotinic acid biosynthesis, as compared to *S. cerevisiae* [6,10]. This fungal pathogen also gained repeated genes, mainly near the telomeres, encoding cell wall proteins, which have an important function in adhesion to different surfaces [6]. The virulence of *C. glabrata* is not well studied, as it behaves quite differently in animals compared to other pathogenic *Candida* species [3]. For example, *C. glabrata* is maintained at high levels in the kidneys of immunocompetent mice during invasive infection, meanwhile resulting in almost no morbidity for the mice [11,12]. As a consequence, *C. glabrata* virulence is usually quantified by fungal burdens in different organs [13,14,15]. Quantifying infections through fungal burden, however, requires a substantial number of animals. To reduce this number, efforts using bioluminescence to study infections have been made mainly for *C. albicans*. For this purpose, a luciferase gene needs to be stably expressed in the fungus, which can then convert a substrate into light [16]. Both the *Gaussia princeps* (copepod) and *Photinus pyralis* (firefly) luciferases have been used successfully in *C. albicans* to image invasive, oral, cutaneous, vaginal, and implant-related biofilm infections [17,18,19,20,21]. The *G. princeps* luciferase has the advantage of being ATP-independent. This facilitates its use on the outside of the cells, for example by attachment to the cell wall, eliminating the need for the substrate to enter the cell [18,20,21]. The firefly luciferase, on the other hand, can be visualized over a much wider spectrum from yellow-green (550 nm) to red (620 nm) wavelengths [22]. A red-shifted firefly luciferase was genetically engineered to have an emission peak at 625 nm, further improving deep-tissue imaging [17]. Both these luciferases could be used to image the reduced fungal burden following antifungal treatment for both invasive and implant-related biofilm infections [17,23]. For *C. glabrata*, however, the use of bioluminescence is not as widespread. Only biofilm-related infections on subcutaneously implanted catheter pieces have been imaged, using a codon-optimized firefly luciferase. This model was used to monitor the activity of different antifungals against biofilms [24,25]. Bioluminescence has, as of yet, not been used to image *C. glabrata* infections affecting internal organs, as often a lower fungal burden compared to *C. albicans* infections is present [3]. Furthermore, stable, high expression of a luciferase gene is more challenging in *C. glabrata,* which is possibly due to its low metabolic flexibility that leads to limited energy availability [26].

A specific type of such infections is urinary tract infections (UTIs). Even though they are mostly associated with *Escherichia coli*, other causative agents including fungi are quite common, especially in infections caused by indwelling catheters, which comprise about 80% of nosocomial UTIs [27]. Catheter-associated fungal UTIs comprise 2–15% of the total infections [27,28,29,30]. Depending on the geographical location, *C. albicans* or *C. glabrata* are the most isolated species in UTIs. *C. glabrata* is the causative agent in 19–48% of cases, with higher incidences occurring due to increased uses of azole antifungals [27,28,29,30,31,32,33,34]. Typically, these infections develop in the bladder but rapidly migrate to the kidneys, resulting in severe symptoms and a mortality up to 30–48%, which is high considering that these patients are not under immunosuppressive therapy [28,31]. Treatment is often difficult, as not all antifungals penetrate well into the bladder. Treatment is even more compromised when catheter use is maintained and biofilms form on the catheter surface, which ultimately results in antifungal resistance [31]. Furthermore, *C. glabrata* seems to be specifically adapted to the urinary tract, as expression of the most important adhesin family, the *EPA* genes, is induced under nicotinic acid limitation, a bladder-specific condition [10]. In this work we further study *C. glabrata* UTIs in a mouse model. In order to reduce the number of animals needed and to make temporal studies possible, we developed a model in which we imaged bladder colonization using bioluminescence, which can be used to determine the effectiveness of antifungal treatment.

## 2. Materials and Methods

### 2.1. Strains, Plasmids, and Growth Media

All plasmid propagations were performed in *Escherichia coli* DH5α, which was grown in LB broth and supplemented with ampicillin (0.1 mg/mL) when necessary. Liquid cultures were grown at 37 °C under constant agitation, and solid plates containing 0.7% Bacto Agar (Difco Laboratories, Basel, Switzerland) were incubated at 37 °C for 16–20 h.

*C. glabrata* and *C. albicans* cells were grown in complete Yeast extract Peptone Dextrose (YPD) medium (2% Bacto Peptone, Difco; 1% yeast extract, Difco and 2% glucose, with or without 2% Bacto Agar, Difco) or Synthetic Complete (SC) medium (0.67% YNB without amino acids, 0.079% CSM complete (MP Biomedicals, Lucerna-Chem AG, Luzern, Switzerland) and 2% glucose) at 30 °C and under continuous agitation for liquid cultures.

*C. glabrata* was transformed using electroporation, and selection was carried out on 200 µg/mL nourseothricin (Werner BioAgents, Jena, Germany).

The red-shifted firefly luciferase gene was amplified via PCR from plasmid pDS1902 [17] using the Phusion high-fidelity Taq polymerase (NEB, Bioconcept AG, Allschwil, Switzerland) (primers LUcopt_BamHI: TGCAGGATCCAAAATGGAAGATGCTAAGA and LUCopt-Rev-KpnI: ATTCGGGTACCACAGCAATTTTACCACC) and cloned into plasmid pVS41 upon restriction digestion with BamHI and KpnI, resulting in plasmid pSSa6. pSV41 is a derivative of pSV20 [35], in which the *SAT1* marker was introduced as a StuI-Pst fragment in pSV20. This plasmid contains a CEN/ARS sequence with a unique HindIII site in the *CEN* sequence, which facilitates integration into the *C. glabrata* genome [36]. A *C. glabrata* codon adapted version of this luciferase was obtained (Eurofins Genomics Germany GmbH, Ebersberg, Germany) and cloned into pVS41, using the BamHI and KpnI restriction sites, resulting in plasmid pSSa13. The *C. glabrata* p*ENO1* and p*TEF1* promoters were amplified from *C. glabrata* genomic DNA (pENO1-SacI+SacII-Fwd: AATTCGGAGCTCCCGCGGGTTTCGAAACAAAGCAGAGTG and pENO1-BamHI-Rev: AATTGCGGATCCTATGATTTATAATATGTGTTTTGTTGTTG; pTEFg-SacI-Fwd: AATTCGGAGCTCCACCGCGGGTTTAGCTATACCAACATGC and pTEFg-BamHI-Rev: AATTCGGGATCCTGCTATATTAGTTCAAGTTAGTTATC) and cloned into pSS13, upon restriction digestion with SacI and BamHI, resulting in plasmids pSSa16 and pSSa23, respectively (plasmids are available at Addgene (Watertown, MA, USA)). The resulting plasmids were digested with HindIII before transformation into *C. glabrata* strain DSY562 [37]. A list of the resulting strains is given in Table 1.

### 2.2. Urinary Tract Infection in Mice

All animal experiments were carried out according to the approval of the Institutional Animal Use Committee, Affaires Vétérinaires du Canton de Vaud, Switzerland (authorizations VD1734.5a and VD2240.2b) at the University Hospital Center of Lausanne. Animals were housed in ventilated cages with ad libidum access to food and water.

At the latest one hour before infection, six-week-old, female Balb/C mice (Charles River, Ecully, France) were injected subcutaneously with 0.1 mg/kg of buprenorphine as a painkiller. Animals were then brought to sleep using isoflurane (3%). The bladder was then emptied, and 2.5 × 10^8^ *C. glabrata* cells or fifty microliters of PBS for the control group was injected directly into the bladder through a catheter (BD Intramedic, 28 mm–61 mm, Becton Dickinson AG, Allschwil, Switzerland). Mice were monitored on a daily basis, including their weight, movement, reflexes, cleanliness, and ability to eat and drink. At three days post infection, mice were sacrificed using CO_2_, and the bladder, kidneys, and spleen were extracted for determination of the fungal burden and/or ELISA. For determination of the fungal burden, organs were homogenized in sterile PBS, and serial dilutions were plated on YPD agar.

### 2.3. Enzyme-Linked Immunosorbent Assay (ELISA)

Organs were homogenized in MPO buffer (200 mM NaCl, 5 mM EDTA, 10 mM TRIS pH 8, 10% glycerol, and one tablet of a complete protease inhibitor cocktail/100 mL (Roche, Basel, Switzerland)) immediately after extraction and centrifuged (1500× *g*, 15 min, 4 °C). The supernatant was collected and stored at −80 °C until utilization. ELISA was carried out according to the specifications of the mouse GM-CSF or IL-1beta ELISA kit (Invitrogen). Results were compared using unpaired t-tests carried out with Graph Prism (Version 9.1.0, GraphPad Software, San Diego, CA, USA).

### 2.4. In Vitro Luminescence Measurement

*C. glabrata* cells were grown overnight in SC medium and were subsequently 10-fold diluted in SC medium. Cells were then further grown until the early exponential phase. Ninety microliters of this culture was transferred to a black, half-area, 96-well plate (COSTAR), and 10 µL of D-luciferin (16 mg/mL) (Biosynth AG, St-Gallen, Switzerland) was injected automatically into each well followed by 3 s of orbital shaking and luminescence measurement using a 96-well plate luminometer (FLUOStar Omega, BMG Labtech, Ortenberg, Germany). The optical density at 600 nm (OD600) of 100 µL of the early exponential phase culture was also measured in the spectrophotometer (FLUOStar Omega) in order to normalize the luminescence by the OD.

### 2.5. In Vivo Bioluminescence and Antifungal Treatment

One day post infection, mice were injected with 100 µL of D-luciferin (18 mg/mL) intraperitoneally and were then anesthetized with isoflurane (3%) and placed in a sealed optical imaging tray ventilated with 2.5% isoflurane. Ten minutes after D-luciferin injection, the optical imaging tray was placed in the Bruker in vivo Xtreme II (Bruker BioSpin MRI GmbH, Ettlingen, Germany) and reconnected to 2.5% isoflurane ventilation. Bioluminescence was measured for 5 min at a field of view 18 × 18 under 8 × 8 binning. Furthermore, an X-ray image (not shown in the present study) and a black and white photograph were also taken, similarly to published procedures [17]. Recording was repeated at three days post infection.

Images were analyzed using the Bruker Xtreme software (Bruker MI SE) by measuring total luminescence in a region of interest (ROI) of identical size for each bladder. Luminescence was expressed as photons/s. In the drug treatment experiment, mice were treated intraperitoneally once daily with 50 mg/kg fluconazole (Diflucan, Pfizer) starting 1 h post infection or with the same volume of PBS for the control animals. Statistical tests were carried out with Graph Prism, using the paired t-test when comparing the treated mice at different timepoints and the unpaired t-test comparing the treated and the untreated group.

## 3. Results

### 3.1. Fate of C. glabrata during Urinary Tract Infections

Because of its clinical relevance, we aimed to further study the *C. glabrata* UTI mouse model used by Vale-Silva et al. [38]. We discovered that not only were the bladder and the kidneys colonized upon inoculation directly into the bladder, but the spleens of more than half of the infected animals were also colonized (Figure 1A). Fungal burden in the spleen was, however, 1–2 log lower compared to the other organs. To reach the spleen, *C. glabrata* has to breach the vascular blood system. It is possible that this fungus uses immune cells as a vector to colonize this organ. We used ELISA to measure two cytokines in infected and non-infected bladders, kidneys, and spleens. Granulocyte macrophage colony stimulating factor (GM-CSF) showed a significantly increased production in infected kidneys compared to non-infected kidneys (unpaired *t*-test *p* = 0.000635) (Figure 1B), while interleukin-1 beta (IL-1beta) was present in significantly higher concentrations in infected spleens versus non-infected spleens (unpaired *t*-test *p* = 0.007897) (Figure 1C). Therefore, the immune system is probably mainly active against the infection in the kidneys and the spleen, as no differences were observed in the measured chemokines of the bladder.

### 3.2. Expression of a Codon Adapted Red-Shifted Luciferase from the ENO1-Promoter Results in High Bioluminescence In Vitro

Since the use of bioluminescence imaging (BLI) can strongly reduce the number of animals needed for in vivo studies, we investigated how we could reach the highest possible luciferase expression in *C. glabrata*. We therefore first tested the red-shifted firefly luciferase (*Ca*LUC), which was used in *C. albicans* by Dorsaz et al. [17] by driving its expression with the *S. cerevisiae PGK1*-promoter (*Sc*p*PGK1*) upon integration into the *C. glabrata* genome. However, we could only detect a low luminescent signal (Figure 2A). We then used a *C. glabrata* codon-adapted version of this luciferase (*Cg*LUC), which significantly increased luminescence. We then further improved luminescence by replacing the *Sc*p*PGK1* by either the *C. glabrata ENO1* or *TEF1* promoters (p*ENO1* and p*TEF1*), which again significantly improved luminescence (Figure 2A). However, luminescence in *C. glabrata* was still approximately 6 times lower compared to the luminescence of the red-shifted firefly luciferase (*Ca*LUC) expressed in *C. albicans* (Figure 2B). We showed that the codon-adapted luciferase gene expressed from the p*ENO1* resulted in the highest luminescence in *C. glabrata* compared to p*TEF1*. Growth of luminescent strains was undistinguishable compared to the wild-type strain in YPD or RPMI at 37 °C (Appendix A).

### 3.3. Expression of a Codon Adapted Red-Shifted Luciferase from the TEF1-Promoter Results in the Highest Luminescence In Vivo

Since the luminescence driven from the p*ENO1* and p*TEF1* constructs was similar, both of them were tested in vivo in the UTI mouse model. The p*TEF1* construct performed significantly better in terms of luminescence in the bladders (unpaired *t*-test, *p* = 0.0075), even though the fungal burden was not different between the two constructs (Figure 3). With the p*TEF1* construct, fungal burdens of more than 200 fungal cells/bladder could be detected using BLI. The correlation between luminescence and the fungal burden, however, was weak (R^2^ = 0.4217), which therefore cannot reflect exact fungal burden quantification using BLI (Figure 4). A linear relationship between absorbance and luminescence can be drawn when cells are grown in vitro (Figure 4).

### 3.4. In Vivo Monitoring of Treatment of a C. glabrata Urinary Tract Infection with Fluconazole

We next aimed to determine whether significant differences in fungal burden, for example because of antifungal treatment, could be detected by BLI. We measured luminescence in fluconazole-treated mice on day one and day three post infection and detected a significant reduction in luminescence on day three compared to day one (Figure 5A). Mice that were treated daily with fluconazole showed significantly lower luminescence on day 3 compared to untreated mice (unpaired *t*-test, *p* = 0.0167). This is consistent with the lower fungal burden in the bladder in treated mice as compared to untreated mice (unpaired *t*-test, *p* < 0.0001). Furthermore, the fungal burden in the kidneys was also significantly decreased in mice treated with fluconazole as compared to untreated mice (unpaired *t*-test, *p* = 0.0007). These data show that the UTI bioluminescence model can be used to study the temporal dynamics of antifungal treatment in *C. glabrata*.

## 4. Discussion

In this work, we investigated the progression of *C. glabrata* urinary tract infections in mice. We observed that the fungus can be detected not only in the bladder and the kidneys but also in the spleen of more than half of the infected animals. Different routes from the bladder and/or kidneys to the spleen are possible. First, *C. glabrata* could penetrate the bladder epithelium to gain access to the bloodstream and reach the spleen via the blood. This is, however, rather unlikely, given that *C. glabrata* has been shown to attach to oral epithelial cells without invading the tissue [39]. So far, no studies have documented invasion of bladder epithelium by *C. glabrata*. One study, however, showed that *C. glabrata* cells can cross the endothelial barrier in human umbilical vein endothelial cells (HUVECs) in vitro [40]. Induced endocytosis of *C. glabrata* to enter into host tissues from the bloodstream has been proposed, but was not formally proven [41]. Another possibility is that *C. glabrata* reaches the spleen hidden inside host immune cells. This yeast was shown to be taken up by neutrophils, monocytes, and macrophages [41,42,43]. Furthermore, both dendritic cells and natural killer cells respond to *C. glabrata*. Dendritic cells were also shown to be required for the persistence of *C. glabrata* in mice [44]. This was attributed to elevated iron levels in the phagolysosome of macrophages, promoting the survival and replication of *C. glabrata* inside these macrophages [45]. *C. glabrata* can also inhibit the acidification of the phagolysosome and, as such, suppress activation of major signaling pathways in host macrophages [46,47,48]. It was recently shown that suppression of IL-1beta production is essential for intracellular survival of macrophages [49]. Production of this cytokine in the spleen could contribute to the low fungal burden in this organ. Even if they are probably not responsible for the transfer of *C. glabrata* from the urinary tract to the spleen, macrophages do seem to be involved in the immune response, since GM-CSF, the macrophage attraction factor, is produced upon infection in the kidneys. This is concordant with other infection models showing that during systemic infection, GM-CSF is produced in the kidneys after 48 h [50] and IL-1beta is induced in lysates of organs in the same model [49]. In the intraperitoneal infection model, GM-CSF was one of the main chemokines produced; however, IL-6 and MCP-1 (monocyte chemoattractant protein) were also produced in this model [51]. In the bladder, we did not observe an increase in the measured chemokines, which might reflect absence of an appreciable inflammatory response to *C. glabrata*, as is the case in the vaginal model [52,53]. However, we only measured two chemokines in the current study, and many others remain to be probed. The transfer of *C. glabrata* from the bladder and kidneys to the spleen remains to be elucidated.

We managed to successfully image *C. glabrata* cells in the bladder; however, we did not consistently manage to image the presence of these fungal cells in the kidneys or spleen. This can possibly be attributed to the lower expression of the luciferase gene in *C. glabrata* compared to *C. albicans*, as well as the lower fungal burden. Although significant progress has been made in bioluminescent imaging of fungi, some limitations still exist. The firefly luciferase that was used here needs oxygen and its substrate, D-luciferin, in order to emit light [16]. It is possible that these co-factors may not reach specific host tissues in the concentration needed for light emission. An alternative to these limitations could be the use of fluorescence imaging. In recent years several red or near-infrared fluorescent proteins were developed in order to enable in vivo imaging [54,55,56]. Fluorescent imaging would eliminate the need for external substrate additions, which can be a limiting factor in some niches of the body. However, excitation of these proteins requires light at wavelengths not absorbed by tissues, given that absorbed light will reduce the extent of excitation. Moreover, the emitted light also has to pass through these tissues to reach camera devices. Therefore, fluorescence imaging has thus far not successfully been used to image deep-seated *Candida* infections in mice. It is possible that in the near future, a shift from bioluminescent to fluorescence in vivo imaging will occur using other fluorescent proteins [57].

We showed that the bioluminescent system presented here can distinguish the differences in fungal burden as a consequence of treatment in the same animal over time. The model, as such, could be used in the future to test the efficacy of novel antifungal molecules as well as possible novel treatment strategies. Combinatorial antifungal therapies have been proposed, but studies have mainly been carried out in vitro; therefore, their value as antifungal therapy is yet to be established [58]. One other advantage of luminescence imaging is that each animal can be evaluated individually, thereby reducing biological variability, since treatment effectivity can be followed inside the same animal over time. Such temporal dynamics of antifungal treatment have already been studied in vivo for *C. glabrata* biofilms on subcutaneously implanted catheter pieces using BLI. The activity of caspofungin against such biofilms resulted in a significant reduction in luminescence after 4–7 days of treatment [24]. In this model, *C. glabrata* does not migrate to any vital organs, and the biofilms are imaged subcutaneously, as such, overcoming the difficulties of light absorption by tissues.

## 5. Conclusions

In conclusion, we have developed a non-invasive model for *C. glabrata* urinary tract infections using bioluminescence. Even though exact quantification of the fungal burden is not possible, our model can be used to test the antifungal activity of new molecules of treatment strategies in vivo in the same animal over time.

## Figures and Tables

**Figure 1 jof-07-00844-f001:**
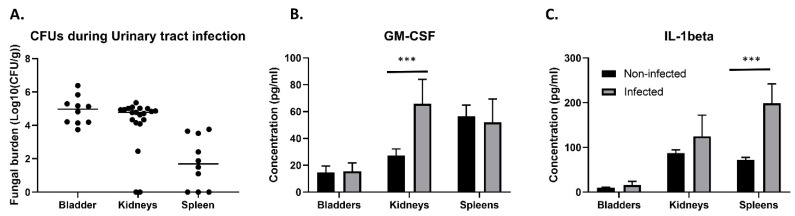
*C. glabrata* colonization of tissues in the UTI model and subsequent immune response. (**A**) Ten mice were challenged with 2.5 × 10^8^ *C. glabrata* cells directly into an emptied bladder. Three days post infection, mice were euthanized, and colony forming units (CFUs) were determined in the bladder, the kidneys, and the spleen. (**B**) Production of GM-CSF; ELISA was carried out for supernatant of homogenized, infected bladders, kidneys, and spleens and their non-infected counter parts. Colonization of the kidneys causes significantly increased production of the macrophage attraction factor GM-CSF (unpaired *t*-test, *p* = 0.000635). (**C**) Colonization of the spleen causes increased production of IL-1beta (unpaired *t*-test, *p* = 0.007897). Error bars show standard deviations. Significance: *** *p* < 0.0001.

**Figure 2 jof-07-00844-f002:**
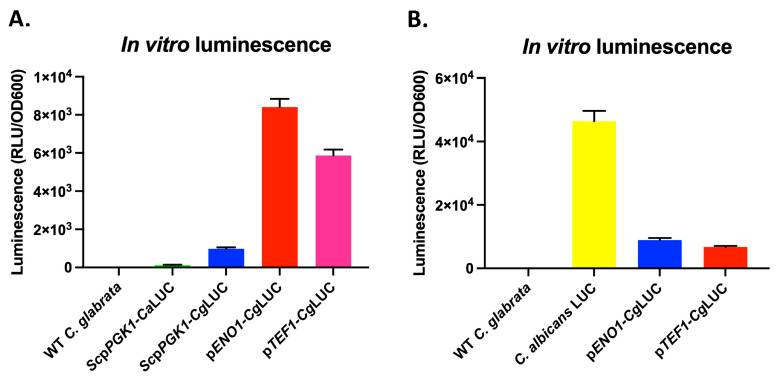
In vitro luminescence of different red-shifted luciferase constructs expressed in *C. glabrata*. Luminescence of in vitro grown early exponential phase cells was measured immediately upon injection of D-luciferin. (**A**) Comparison of the non-codon-adapted luciferase (*Ca*LUC) to the codon-adapted luciferase (*Cg*LUC) expressed from either the *S. cerevisiae PGK1* promoter (*Sc*p*PGK1*), the *C. glabrata ENO1* promoter (p*ENO1*), or the *C. glabrata TEF1* promoter (p*TEF1*). (**B**) A comparison between the two best constructs in *C. glabrata* and the *C. albicans* luminescent strain (*C. albicans* LUC) used by Dorsaz et al. [17]. All error bars show standard deviations.

**Figure 3 jof-07-00844-f003:**
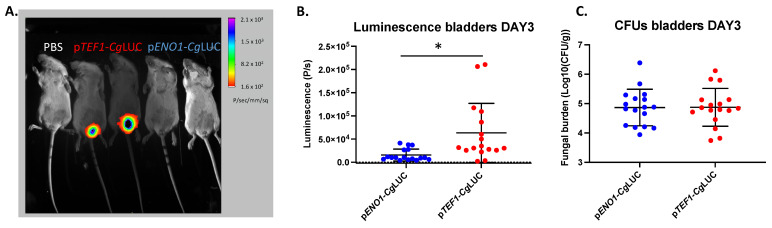
In vivo comparison of two luciferase-expressing constructs in *C. glabrata* in the bladder. (**A**) BLI of infected bladders with either the p*TEF1*-*Cg*LUC construct or the p*ENO1*-*Cg*LUC construct. (**B**) BLI in the bladder for the two constructs. Seventeen mice for each group were infected with 2.5 × 10^8^ *C. glabrata* cells directly into the bladder. Luminescence in the bladders was measured on day three post infection. A statistically significant, higher luminescence can be observed in mice infected with *C. glabrata* containing p*TEF1*-*Cg*LUC (unpaired *t*-test, *p* = 0.075). (**C**) Fungal burden in the bladders of mice infected with *C. glabrata* expressing different luminescence constructs. All error bars show standard deviations. Significance: * *p* < 0.05.

**Figure 4 jof-07-00844-f004:**
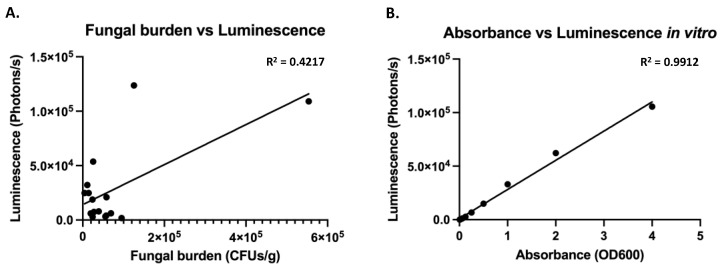
Luminescence does not correlate perfectly to fungal burden in vivo, even though good correlation exists in vitro (**A**) In vivo luminescence in the bladder was correlated to fungal burden using linear regression in Graph Prism. (**B**) In vitro measured luminescence was correlated to absorbance using linear regression in Graph Prism.

**Figure 5 jof-07-00844-f005:**
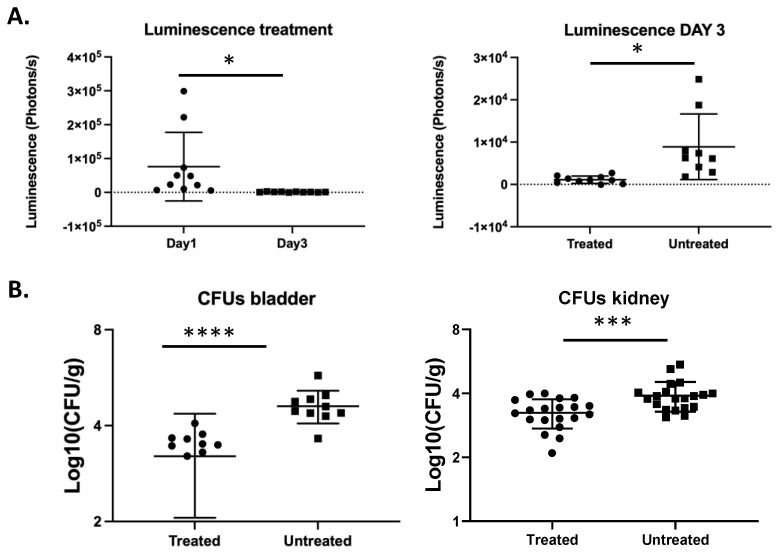
Bioluminescence imaging reflects the decreased fungal burden because of antifungal treatment. Ten mice per group were infected with 2.5 × 10^8^ *C. glabrata* cells directly into the bladder. The treated group was injected intraperitoneally with fluconazole (50 mg/mL) on a daily basis. Luminescence in the bladders was measured on day one and three post infection. (**A**) Luminescence in the bladders upon fluconazole treatment (paired *t*-test, *p* = 0.0439) (**left**). Luminescence in the bladders of treated mice is significantly lower compared to untreated mice (unpaired *t*-test, *p* = 0.0167) (**right**). (**B**) Fungal burden in the bladders and the kidneys upon fluconazole treatment (unpaired *t*-test, *p* < 0.0001 and *p* = 0.0007, respectively). All error bars show standard deviations. Significance: **** *p* < 0.0001, *** *p* < 0.001, * *p* < 0.05.

**Table 1 jof-07-00844-t001:** List of *C. glabrata* and *C. albicans* strains used in this work.

Strain Name	Parent Isolate	Incorporated Plasmid	Reference
DSY562	none	none	[37]
DSY4823	CAF4-2	pDS1902	[17]
SSY3	DSY562	pSSa6	This work
SSY5	DSY562	pSSa13	This work
SSY8	DSY562	pSSa16	This work
SSY21	DSY562	pSSa23	This work

## Data Availability

The datasets generated during and/or analyzed during the current study are available from the corresponding author upon request.

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
