# Peer review of "Investigating Candida glabrata Urinary Tract Infections (UTIs) in Mice Using Bioluminescence Imaging"

_jof, 2021, doi:10.3390/jof7100844_

Round 1

Reviewer 1 Report

In this short, mostly technical work, the authors introduce several luminescent C. glabrata strains and test their potential use as a tool in studying fungal infections, to mixed results. I did like the idea, and the critical evaluation e.g. of the correlation luminescence/cfu. While the data set is somewhat limited, the study also gives promising first results that these strains may complement infection models in the future and help to reduce the number of mice required in experiments. Some more tests could have been done (see comments below), but even as it is, these tools will be of great interest to scientists in the field.

There are some minor things to consider:

  • The correlation diagram in fig. 4 assumes a direct, linear correlation between detected luminescence and cfu. With cells potentially clumping, covering each other, etc. this may not be necessarily true. Have the in vitro assays in e.g. fig 2 maybe been performed with different cfu of C. glabrata (best in stationary conditions), and is the correlation luminescence/cfu linear in these cases?
  • How has the insertion site of the construct been checked? And where did the construct insert in the genome? This should be stated in the M&M or results section, as it may be important for future uses of the strains.
  • The discussion focuses a lot on the potential "hitchhiking" of C. glabrata inside of macrophages. While interesting, this is based on very little data from this manuscript, which focused mainly on the suitability of luminescent strains in infection models. A matter of taste, surely, but maybe some of the more speculative parts could be toned down based on the limited data on this specific subject?

Some minor problems I found are:

  • line 41: "efforts .. have occured"?
  • line 60: How is stable high expression related to metabolic flexibility?
  • line 148: Where were the X-ray images used?
  • line 157 (and following): C. glabrata is not italicized
  • fig. 1: "Kindeys" is a typo in the graph (B) axis label
  • fig. 4: R² value is given with decimal comma instead of point (see also line 212 for R² vs R^2)
  • line 292: is it not rather wavelengths that _are_ absorbed that may cause problems?

Author Response

We thank reviewer 1 for constructive comments.

(1) The correlation diagram in fig. 4 assumes a direct, linear correlation between detected luminescence and cfu. With cells potentially clumping, covering each other, etc. this may not be necessarily true. Have the in vitroassays in e.g. fig 2 maybe been performed with different cfu of  glabrata (best in stationary conditions), and is the correlation luminescence/cfu linear in these cases?

Answer: Another panel was added to Fig 4 which shows a linear correlation between luminescence and in vitro grown C. glabrata measured by optical density.

(2) How has the insertion site of the construct been checked? And where did the construct insert in the genome? This should be stated in the M&M or results section, as it may be important for future uses of the strains.

Answer: A comment was added on the integrative plasmid (L 101-103):

“This plasmid contains a CEN/ARS sequence with a unique HindIII site in the CEN se-quence which facilitates integration into the C. glabrata genome [36].”

(3) The discussion focuses a lot on the potential "hitchhiking" of  glabratainside of macrophages. While interesting, this is based on very little data from this manuscript, which focused mainly on the suitability of luminescent strains in infection models. A matter of taste, surely, but maybe some of the more speculative parts could be toned down based on the limited data on this specific subject?

Answer: We removed the speculative statements and replace them by (L. 274-275):

“Production of this cytokine in the spleen could contribute to the low fungal burden in this organ”.

We also added a sentence at the end of praragraph (L.286-287):

“….and the transfer of C. glabrata from the bladder and kidneys to the spleen, however, remains to be elucidated”.

(4) line 41: "efforts .. have occured"? à

Answer: changed to “…efforts have been made”

(5) line 60: How is stable high expression related to metabolic flexibility?

Answer: we wrote (L 59-61)

“Furthermore, stable high expression of a luciferase gene is more challenging in C. glabrata, which is possibly due to its low metabolic flexibility that leads to limited energy availability [26].”

(6) line 148: Where were the X-ray images used?

Answer: We did not include these images in the current manuscript. We modified the sentence Lane 155-157:

“Furthermore, an X-ray image (not shown in the present study) and a black and white photograph were also taken, similarly to published procedures [17].

(7) line 157 (and following): C. glabrata is not italicized

Answer: corrected

(8) "Kindeys" is a typo in the graph (B) axis label

Answer: corrected

(9) 4: R² value is given with decimal comma instead of point (see also line 212 for R² vs R^2)

Answer: corrected

(10) line 292: is it not rather wavelengths that _are_ absorbed that may cause problems?

Answer: we modidified the sentence (L299-301):

“However, excitation of these proteins requires light at wavelengths not absorbed by tissues, given that absorbed light will reduce the extent of excitation.”

Reviewer 2 Report

In this manuscript, the authors evaluate the tracking of Candida glabrata during urinary tract infections in mice using bioluminescence imaging. Animal model using genetically modified strains was also used to monitor  fluconazole treatment effectiveness. The subject seems to be interesting and most of the results and methods are readily understandable. However, some elements of the work need to be clarified or revised.

Major comments

A part of the discussion from line 263 to267 „These increased IL-1beta levels, could also suggest that macrophages containing yeast cells are transported to the spleen, where macrophages manage to combat C. glabrata infection through IL-1beta production, leading to only low fungal burdens” is not clear or is incorrect, since macrophages are typical tissue-resident cells not able to migrate by blood. Only one of the macrophage precursors – monocytes are present in blood ! Actually I do not understand this explanation.

The method section lacks a description of the statistical analyses and it is not specified what type of error bars (SD or SEM) are shown in the graphs. Refers to Fig. 1, 2, 3B and C, and 5. It must be added in the description of the Figures.

The statistical analysis of the data presented in Figure 5A is incorrect -  In this case the paired test should be performer as analyzed is the same set of variables before and after treatment.

Minor comments

A working desscription of type 2.5*10^8 should be corrected. Applies to the entire work.

If 17 mice were used for the experiments shown in Fig. 3B, why are there only 14 points (referring to these mice)  in Fig. 3 C?

The letter C is missing in description of Fig. 3

Author Response

We thank reviewer 2 for constructive comments

Major comments

(1) A part of the discussion from line 263 to267 „These increased IL-1beta levels, could also suggest that macrophages containing yeast cells are transported to the spleen, where macrophages manage to combat C. glabrata infection through IL-1beta production, leading to only low fungal burdens” is not clear or is incorrect, since macrophages are typical tissue-resident cells not able to migrate by blood. Only one of the macrophage precursors – monocytes are present in blood ! Actually I do not understand this explanation.

Answer: As asked by reviewer1, we removed the speculative statements and replace them by (L. 274-285):

“Production of this cytokine in the spleen could contribute to the low fungal burden in this organ”.

We also added a sentence at the end of praragraph (L.286-287):

“….and the transfer of C. glabrata from the bladder and kidneys to the spleen, however, remains to be elucidated”.

(2) The method section lacks a description of the statistical analyses and it is not specified what type of error bars (SD or SEM) are shown in the graphs. Refers to Fig. 1, 2, 3B and C, and 5. It must be added in the description of the Figures.

Answer: We stated in each figure legend that error bars show the standard deviation. In the materials and methods section, we mentionedd that statistical tests (t-tests paired or unpaired) were performed  wiht Graph Prism.

(3) The statistical analysis of the data presented in Figure 5A is incorrect -  In this case the paired test should be performer as analyzed is the same set of variables before and after treatment.

Answer: This was corrected in Figure 5 legend to a paired t-test

(4) A working desscription of type 2.5*10^8 should be corrected. Applies to the entire work.

Answer: This was changed to 2.5x108 and where it applies.

(5) If 17 mice were used for the experiments shown in Fig. 3B, why are there only 14 points (referring to these mice)  in Fig. 3 C?

Answer: We corrected the number of data points showed in the Figure due to incomplete transfer of data from a source Excel file.

(6) The letter C is missing in description of Fig. 3

Answer: corrected.